# The Tumor Microenvironment in Colorectal Cancer Therapy

**DOI:** 10.3390/cancers11081172

**Published:** 2019-08-14

**Authors:** Leire Pedrosa, Francis Esposito, Timothy M. Thomson, Joan Maurel

**Affiliations:** 1Department of Medical Oncology, Hospital Clinic of Barcelona, 08036 Barcelona, Spain; 2Translational Genomics and Targeted Therapeutics in Solid Tumors Group, Institut d’Investigació Biomèdica August Pi i Sunyer (IDIBAPS), 08036 Barcelona, Spain; 3Department of Medicine, University of Barcelona, 08036 Barcelona, Spain; 4Networking Center for Biomedical Research in Hepatic and Digestive Diseases (CIBER-EHD), Instituto de Salud Carlos III, 28029 Madrid, Spain; 5Laboratory of Cell Signaling and Cancer, Institute for Molecular Biology, Science Research Council (CSIC), 08036 Barcelona, Spain

**Keywords:** colorectal cancer, mCRC, tumor microenvironment (TME), tumor-infiltrating lymphocytes (TILs), angiogenesis, tumor metabolism

## Abstract

The current standard-of-care for metastatic colorectal cancer (mCRC) includes chemotherapy and anti-angiogenic or anti-epidermal growth factor receptor (EGFR) monoclonal antibodies, even though the addition of anti-angiogenic agents to backbone chemotherapy provides little benefit for overall survival. Since the approval of anti-angiogenic monoclonal antibodies bevacizumab and aflibercept, for the management of mCRC over a decade ago, extensive efforts have been devoted to discovering predictive factors of the anti-angiogenic response, unsuccessfully. Recent evidence has suggested a potential correlation between angiogenesis and immune phenotypes associated with colorectal cancer. Here, we review evidence of interactions between tumor angiogenesis, the immune microenvironment, and metabolic reprogramming. More specifically, we will highlight such interactions as inferred from our novel immune-metabolic (IM) signature, which groups mCRC into three distinct clusters, namely inflamed-stromal-dependent (IM Cluster 1), inflamed-non stromal-dependent (IM Cluster 2), and non-inflamed or cold (IM Cluster 3), and discuss the merits of the IM classification as a guide to new immune-metabolic combinatorial therapeutic strategies in mCRC.

## 1. Cancer: Immune Surveillance, Escape, and Immunotherapy

The immune surveillance hypothesis of cancer was formulated at the beginning of the 20th century by Paul Ehrlich, who proposed that the immune system is key to eradicating undetectable cancer cells that appear at a high frequency, and that failure of this immune surveillance leads to the emergence of clinically detectable tumors [1]. This hypothesis has been overwhelmingly validated since its original formulation through multiple lines of evidence. Later on, Ehrlich’s hypothesis was complemented by Robert Schreiber’s theory of immunoediting [2], whereby the immune system exerts a selective pressure that favors the outgrowth of poorly immunogenic tumor clones that escape immune recognition. This notion has also been validated through comprehensive genomic and transcriptomic analyses that have shown that tumor heterogeneity and evolution are molded through immune recognition and editing [3].

However, until relatively recent times, effective approaches to reinvigorate the immune recognition of tumors and its oncolytic activity were limited to a handful of tumor types in apparently idiosyncratic settings. A significant advance in systematic tumor immunotherapy was the development of chimeric antigen receptor (CAR) T cells [4]. Chimeric antigen receptors are genetically engineered receptors that couple a single-chain Fv domain recognizing a pre-defined cell-surface antigen to intracellular T-cell signaling domains of the T-cell receptor, thereby redirecting cytotoxic T lymphocytes to cells expressing the target antigen. This approach has been particularly successful in the management of leukemias with the clonal expansion of neoplastic cells that bear well-defined differentiation antigens, such as CD19 on chronic lymphocytic leukemia (CLL) and B-cell acute lymphoblastic leukemia (B-ALL), which are eradicated upon CD19-specific CAR-T cell targeting.

A further revolution in immunotherapy was brought about by the demonstration that T cells are susceptible to suppressive (checkpoint) signals and that they can be reactivated by blocking checkpoint signaling [5,6]. The CTLA-4 or PD-1 (or their ligands) immune checkpoint blockade (ICB) successfully reactivates anti-tumor immune effector functions in significant subsets of melanoma and other cancers, including lung, colorectal, gastric, or hepatocellular cancer, frequently with impressive responses, prompting enthusiasm for this approach with a consequent flurry of clinical trials for its application to many other types of cancer [7], either as stand-alone or in combinatorial therapeutic schemes [8]. However, despite these successes, a large proportion of patients do not benefit from CBI and a significant fraction develop immunoresistance [9].

The clinical response to ICB has been shown to correlate with (1) the mutational load (non-synonymous/frameshift mutations) of tumors, presumably associated with the expression of neoantigens; (2) expression of PD-L1 by tumor cells (cell membrane-associated ligand of the PD-1 checkpoint receptor); and (3) the abundance of circulating Ki67^+^CD8^+^ T cells relative to the tumor burden. These observations point to some of the mechanisms by which tumors escape CB reactivation immunotherapy after the initial response. These include the emergence of poorly immunogenic tumor clones with a low neoantigen burden; the downregulation of major histocompatibility complex (MHC) molecules on tumor cells, thus curtailing neoantigen presentation [10]; or the upregulation of PD-L1 and PD-1 through the activation of IFN-γ (concomitant with T*_eff_* cell activation) [11]. Additional mechanisms of acquired immunoresistance include T*_eff_* and/or natural killer (NK) cell exhaustion [12]; the recruitment of myeloid-derived suppressor cells to the tumor microenvironment [13]; and immunosuppression induced by tumor-generated metabolic microenvironments, including nutrient exhaustion (including glucose, glutamine, or tryptophan) caused by tumor cell metabolism [14], hypoxia, or enhanced acidosis by the accumulation of lactate produced by tumor cells [15]. A distinct mechanistic class of acquired cancer immunoresistance is the modulation of immune subsets and functions through gut microbiome interactions [16].

## 2. Metabolic Reprogramming in Tumor Cells and Associated Immune Cells

Tumor cells undergo metabolic reprogramming, essential for the maintenance of malignant features and fitness, particularly in dynamic environments [17]. In addition to the well-known switch to aerobic glycolysis, by which tumor cells favor glycolysis over mitochondrial metabolism for the generation of adenosine triphosphate (ATP) (Warburg effect) [18], tumor cells resort to alternative energy production pathways and carbon sources when exposed to environmental stress, such as hypoxia or drugs. Tumor hypoxia promotes the stabilization and activation of hypoxia-inducible factor 1alpha (HIF-1α), which facilitates the adaptation of tumors to hypoxic stress by modulating tumor cell metabolism, survival, angiogenesis, migration, invasion, and metastasis and, eventually, chemoresistance and radiation resistance [19]. Indeed, high HIF-1α expression is associated with a poor prognosis in many cancers, including CRC [20]. Major effects of HIF-1α activation on metabolism are stimulation of glycolytic energy production by promoting the expression of the glucose transporter GLUT1 (SLC2A1) and glycolytic enzymes and downregulation of mitochondrial oxidative phosphorylation (OXPHOS) by promoting the expression of pyruvate dehydrogenase kinase 1 (PDK1) [21]. The combination of blunted mitochondrial function and high glycolytic activity associated with hypoxia and HIF-1α activity leads to the accumulation of cytoplasmic pyruvate and nicotinamide adenine dinucleotide (NADH). In order to dispose of these compounds, lactate dehydrogenase A (LDHA) is induced by HIF-1α and catalyzes the conversion of pyruvate and NADH to lactate and NAD+ [22], after which lactate is exported to the extracellular milieu through the HIF-inducible plasma membrane monocarboxylate transporter 4 (MCT4, SLC16A4). Therefore, a collateral effect of these processes is acidosis of the extracellular environment surrounding tumor cells under hypoxia.

Upon activation, T cells undergo rapid metabolic reprogramming that shifts the fatty acid oxidation of quiescent states to the augmented uptake and use of nutrients in order to generate the energy and cellular resources necessary to meet the new cellular states [23]. Importantly, distinct T cell subsets exhibit different metabolic profiles, at naïve, activated, memory, or exhausted states [24]. Effector T cells (T*_eff_*) are highly glycolytic, while suppressive regulatory T cells (T*_reg_*) display a mixed metabolism with enhanced lipid oxidation. The mechanistic target of rapamycin (mTOR) complex 1 (mTORC1) is essential for the differentiation of naïve T cells into helper T_H_1 and T_H_17 cells, the cytolytic activity of CD8^+^ memory T cells, and the generation of T*_eff_*, while opposing the generation and function of T*_reg_* cells. On the other hand, mTORC2 promotes the differentiation of naïve CD4^+^ T cells into T_H_2 cells [25]. Therefore, the inhibition of mTOR blocks the activation and effector functions of T cells and favors the generation of T*_reg_*. Consistently, the impaired uptake of nutrients, including glucose, leucine, or glutamine, as observed in tumor microenvironments, inhibits the generation of T*_eff_*, but not T*_reg_*. Moreover, glutamine metabolism is essential for the generation of T_H_1 cells [26]. Remarkably, while glutamine uptake is required for the generation and function of T_H_1 and T_H_17 T*_eff_* cells, glutaminolysis through glutaminase (GLS) activity is required for T_H_17 cells, while it is inhibitory for T_H_1 cells [27]. This suggests that targeting GLS should concomitantly inhibit inflammatory T_H_17 cells and enhance T_H_1 functions. Similarly, natural killer (NK) cells, while requiring glutamine uptake, are not affected by GLS inhibition [28]. This presents the exciting possibility of tackling glutaminolysis in order to inhibit GLS-dependent tumor cells while preserving T_H_1 and NK cell functions with potential anti-tumoral activity.

## 3. Metastatic Colorectal Cancer

Colorectal cancer (CRC) is one of the most frequent cancers worldwide and the second leading cause of cancer-related death in Western countries. In the last two decades, the use of chemotherapy (CHT) and targeted-agents (TA) has afforded significant progress in survival [29,30]. However, in spite of high initial responses (>50% risk ratio (RR)), time to tumor progression ranges between 9 and 12 months and practically all patients with metastatic CRC (mCRC) succumb to disease progression.

CRC tumors with microsatellite instability, or that are mismatch repair-deficient (MSI), have a high mutational burden (particularly frame-shift mutations) that creates many neoantigens that are loaded on the MHC of antigen-presenting cells and recognized as foreign by T cells [31]. As a consequence, MSI tumors are characterized by a microenvironment rich in tumor-infiltrating lymphocytes (TIL) and high levels of type-I interferons in comparison with MSS. The strong activation of tumor-directed immune cells triggers the feedback expression of immune checkpoint blockade receptors and ligands, such as PD-1 and PD-L1, on tumor cells, TILs, and tumor-associated macrophages (TAMs). This explains why patients with MSI mCRC constitute a rare group (5% of all cases) responsive to ICB (PD-1 blockade) [32]. In 2017, ICB (pembrolizumab and nivolumab) received regulatory approval by the FDA for the management of MSI mCRC as second-line therapy after first-line chemotherapy, with or without TA failure [33,34,35]. The response rates in phase II studies range between 30% and 54%. The tumor mutational burden (TMB) can potentially discriminate optimal CBI responders in MSI mCRC patients [36]. Unfortunately, the vast majority of mCRC patients (95%) are microsatellite stable (MSS) with low TMB and a lack of immune cell infiltration, and thus do not respond to PD-1 or PD-L1 inhibitors [32].

## 4. Reprogramming of CRC in Response to Chemo- or Radio-Therapy: Angiogenesis, Immune Response, and Metabolism

When designing therapeutic strategies in cancer, it is critical to consider the multidimensionality of the expected responses and all possible scenarios that may be predictive of a responsive or unresponsive tumor environment. For example, single high-dose chemotherapy or irradiation induces endothelial cell apoptosis and senescence via increased ALK5 and sphingomyelinase expression [37]. This causes vessel regression and vascular collapse, which is accompanied by reduced perfusion, eventually resulting in tissue hypoxia, which leads to a vascular rebound effect by growth factor-induced vasculogenesis and angiogenesis [38]. This promotes different endothelial cell functions that result in vascular growth induction and enhanced tissue perfusion. Both the vascular rebound effect and vascular growth induction provide opportunities for therapeutic intervention in combination with chemotherapy or radiotherapy [39].

Chemotherapy or irradiation of tumor cells can induce the expression of interferon beta (IFN-β) through cytosolic double-stranded (ds) DNA/guanosine monophosphate (GMP)-AMP synthase/stimulator of interferon genes (dsDNA/cGAS/STING) signaling [40]. In parallel, chemotherapy and irradiation promote an immune response through various mechanisms. Upon chemo- or radiotherapy-induced cell death, damage-associated molecular patterns (DAMPs) are released and activate antigen presenting cells, including dendritic cells and stromal fibroblasts [41], through pattern-recognition receptors (PPR) [42]. Further, cytosolic DNA and DAMPS induce the activation and assembly of AIM2 and NLRP3 inflammasomes, respectively, which canonically leads to caspase-1-mediated maturation and the release of IL-1β and IL-18 through exocytosis or gasdermin-dependent pyroptosis [43] from tumor microenvironment components including tumor cells, stromal fibroblasts, TILs, and endothelial cells [44].

IL-1 orchestrates a plethora of effects [45]. When liganded to its cognate receptor on T cells (ncTh1, Th17, CD8, Tγδ17), IL-1β induces the expression and secretion of IFN-γ and IL-17, with consequent local amplification of innate and adaptive immunity, but also additional effects. On one hand, IFN-γ induces the expression PD-L1 of on tumor cells [46]. On the other hand, it stimulates the expression of leukocyte adhesion molecules in the vessel wall, which contributes to increased immune cell recruitment. Vessel regression induces hypoxia which increases the expression of growth factors and chemokines that affect immune cell recruitment and polarization [47]. Finally, chemo/radiotherapy induces the expression of molecules on the tumor cell surface, like MHC-I and Fas [48,49], which increases tumor cell killing by immune cells. Therefore, targeting immune suppressive mechanisms provides opportunities for therapeutic intervention in combination with chemo/radiotherapy.

## 5. Immune Microenvironments in CRC

Given the mechanistic importance of PD-L1 expression in anti-tumor ICB, its detection as a predictive biomarker has justifiably garnered much attention. However, its detection by immunohistochemistry (IHC) is confounded by multiple issues (e.g., variable detection antibodies, differing IHC cutoffs, tissue preparation, and processing variability) and the identities of the various cell types being stained (e.g., tumor or myeloid cells). Emerging data suggest that patients whose tumors (e.g., lung cancer or melanoma) over-express PD-L1 by IHC show improved clinical outcomes in response to ICB. In mCRC, PD-L1-positive expression in tumor cells ranges between 22% and 38% in MSI and 13% and 67% in MSS [35,50,51,52]. In contrast, no correlation was found between PD-L1 positivity (>1% positivity) and ICB efficacy in MSI-H [34,35]. In fact, it seems that the expression of PD-L1 in MSI-H tumors is confined to polarized macrophages (CD163^+^) at the invasive front and in the stroma, and is not found on tumor cells. he TIL density (CD3^+^, CD4^+^, CD8^+^, and FOXP3^+^ T*_reg_*) is prominent in MSI tumors relative to MSS tumors [31,53] and CD8^+^ and CD3^+^, but not FOXP3, and TIL densities correlate well with the percentage of frameshift mutations detected in MSI tumors [54]. Among MSI mCRC cases, tumors in Lynch-associated patients show increased TIL infiltration (CD3^+^, CD8^+^, or T*_reg_*) and higher TMB (median 384 vs. 71) compared to sporadic patients [55]. The quantification of T cells and cytotoxic T cells (CD3 and CD8) in mCRC tumors (immunoscore) shows that a higher immunoscore correlates with a decreased likelihood of metastasis [56] and can also predict overall survival [57]. However, it excludes regulatory T cells (T*_reg_*) and inflammatory T cells (IL-17^+^), which have potentially important roles in CRC immunosuppression [58].

Tumor-associated macrophages (TAMs) also play fundamental roles in remodeling the tumor immune microenvironment. Macrophages differentiate into two main phenotypes: the classically activated phenotype or M1 macrophages, displaying pro-inflammatory and tumor suppressive roles through the induction of type-I helper T-cell (T_H_1) responses, and the alternatively activated phenotype, or M2 macrophages, involved in type-II helper T-cell (T_H_2) responses, with both immunosuppressive and tumor-promoting (pro-angiogenic) activities. Enhanced infiltration with M1-macrophages at the tumor front has recently been correlated with a better prognosis in CRC patients. However, the majority of TAMs display the M2 phenotype. Given that multiple subpopulations of TAMs exist within a tumor, a promising field of research is focused on interrogating the temporal evolution of TAM heterogeneity, including interactions and topographical relationships in the context of changing tumor microenvironments over the course of tumor progression, including under hypoxia and drug pressure.

As pointed out above, cancer metabolism and its tight association with the microenvironment have been recognized in the last decade as an emerging hallmark to understand immune efficacy [59]. Indeed, it is increasingly being realized that the activities of different immune cells, including T cells, NK cells, or macrophages, may be better predicted as a function of their specific metabolic programs than by their phenotypes, as defined through differentiation markers [60]. As outlined above, each type of CD4^+^ T helper cell preferentially utilizes a source of energy production, with naïve and regulatory T cells utilizing fatty acid oxidization (FAO) as a main source of energy production and effector T helper cells (T_H_1, T_H_2, and T_H_17) favoring glycolysis. In fact, the balance between glycolitic and lipid oxidative metabolism in cancer cells determines the ratio of associated T*_reg_*, T*_eff_*, and T_H_17 cells [24].

## 6. The GAIS-42 Immune-Metabolic Classification of mCRC

The foregoing considerations underscore the need for new treatments and combinational strategies in the management of mCRC, which require the parallel identification of appropriate predictive biomarkers. We and others have applied gene expression-based immune-related signatures coupled to TMB scores in order to better capture the complexity of the tumor-immune interactions and predict ICB efficacy [61,62,63,64,65].

None of the existing CRC immune signatures [66,67,68] have proven to be a useful guide to design ICB in combination with additional therapies to broaden the range of patients that may benefit from such new strategies. This may stem from shortcomings in integrating immune signatures with information on the functional microenvironment status and also metabolic reprogramming impacting tumor and non-tumor cells. Recently, we conducted an analysis of 100 consecutive mCRC patients in which Next Generation Sequencing (NGS) data yielding immune signatures were integrated with TMB, clinical scores, and metabolic pathway inferences. The resulting GAstrointestinal Immune-Signature (GAIS-42-patent P5020EP00) contained 42 genes, selected on the basis of published pre-clinical research and the IO 360 Immune signature (IO360), which contains 770 immune-related genes and classifies mCRC into three distinct immune-metabolic clusters, namely, inflamed-stromal-dependent (IM Cluster 1), inflamed-non stromal-dependent (IM Cluster 2), and non-inflamed/cold (IM Cluster 3) [69]. Below, we provide a detailed description of these clusters, including inferences on tumor-associated immune profiles and metabolic reprogramming, as well as proposals for cluster-specific therapeutic approaches (Table 1). Briefly, the clusters are designated inflamed-stromal-dependent (IM Cluster 1) if they express cancer-associated fibroblast (CAF) markers and appear enriched in myeloid-derived suppressor cells (MDSC), M2, and T*_reg_*. The inflamed-non stromal-dependent (IM Cluster 2) cluster is characterized by a distinct MDSC, M2, and T*_reg_* enrichment, but without CAF markers. Finally, the non-inflamed/cold (IM Cluster 3) cluster is characterized by the absence of MDSC, M2, T*_reg_*, and CAF markers.

### 6.1. Cluster 1: Inflamed-Stromal Dependent

#### 6.1.1. Metabolism and Microenvironment

Cluster 1 encompasses 35–40% of mCRC patients and is characterized by the up-regulation of cancer-associated fibroblast (CAF)-related genes, and a relatively modest immune infiltrate and weaker immune checkpoint activation, compared to Cluster 2 tumors. Its metabolic transcriptomic signature suggests a bypass of mitochondrial entry of pyruvate (due to high PDK1 levels, inhibitory of PDH), with concomitant derivation to lactate (high LDH-A), accompanied by an enhanced uptake of glutamine (high levels of the glutamine transporter SLC1A5), which would be required for anaplerotic replenishment of the tricarboxylic acid cycle (TCA) cycle. This scenario is suggestive of an ongoing Warburg effect, with lactate-mediated tumor microenvironment (TME) acidification. 

The expression profile of metabolic genes in Cluster 1 suggests a predominance of an aerobic glycolytic pathway over the oxidative mitochondrial TCA cycle. A hypoxic environment would be consistent with this metabolic state, as HIF-1α induces an upregulation of PDK1 (Figure 1). Additionally, specific oncogenic mutations and an active Wingless-Int-1 (Wnt) pathway could contribute to this metabolic phenotype in Cluster 1 tumors. 

In terms of metabolic gene profiling, Clusters 1 and 3 share the overexpression of MCT1 (SLC16A1, predominantly lactate importer), PDK1, and LDH-A. A major difference between these two clusters is the enrichment in Cluster 1 in genes associated with cancer-associated fibroblasts (CAFs), M2-polarized macrophages, and polymorphonuclear (granulocyte) myeloid-derived suppressor cells (gMDSC) (Table 1). Granulocytes (gMDSC) produce ROS, including hydrogen peroxide, which generates a pseudo-hypoxic TME [70,71], which contributes to a hypoxic phenotype in both tumor cells and CAFs. Lactate effluxed from the latter might be imported into cancer cells through MCT1 and used to generate pyruvate through the activity of LDH-B, which is also highly expressed in Cluster 1.

Mutated Kirsten rat sarcoma viral oncogene homolog (KRAS) and v-Raf murine sarcoma viral oncogene homolog B (BRAF) contribute to a glycolytic reprogramming that fuels the conversion of glucose to lactate [72]. For instance, mutant KRAS in pancreatic cancer has been shown to activate glycolytic and hexosamine biosynthesis and a non-oxidative branch of the pentose phosphate pathways [73]. Moreover, in pancreatic cancer, mutant KRAS reprograms the use of glutamine-derived glutamate, such that instead of using glutamate dehydrogenase (GLUD1) to convert glutamate into α-ketoglutarate and TCA entry, glutamine-derived aspartate is converted into oxaloacetate by mutant KRAS-dependent upregulated GOT1 in the cytoplasm [74]. The resulting oxaloacetate can be converted into malate and then pyruvate, which in turn can be converted into lactate through the action of LDH-A, thus increasing TME acidification.

Aberrant WNT signaling is also linked to the metabolic reprogramming of cancer cells with the constitutive activation of this pathway. Nuclear β-catenin favors aerobic glycolysis through the transcriptional activation of v-MYC avian myelocytomatosis viral oncogene homolog (MYC) and PDK1, and lactate production through the upregulation of LDH-A [75,76]. Interestingly, 35% of cases in Cluster 1 harbor *BRAF* mutations, in contrast to 20% in each of Clusters 2 and 3. Therefore, 60% of *BRAF* mutations occur in Cluster 1 tumors. Although mCRC patients with *BRAF* mutations have a lower frequency of *APC* mutations compared to mutant *RAS* mCRC patients, alternative pathways can be used to achieve a constitutive activation of the Wnt-β-catenin pathway. Therefore, *CDX2* loss [77,78], and DNA hypermethylation of *AXIN2*, associated with mutant BRAF, contribute to enhanced β-catenin/TCF/MYC transcriptional activity. Tumors harboring mutant BRAF also show overexpression of the small ubiquitin-like modifier (SUMO) protein ligases PIAS 1 and PIAS 3 (protein inhibitor of activated STAT), which may potentiate epithelial-mesenchymal transition [79]. Finally, 35% of cases in Cluster 1 lack mutations in *RAS* or *BRAF*.

Cluster 1 overexpresses classical CAF markers, such as alpha-smooth muscle actin (a-SMA), fibroblast activation protein (FAP), and platelet-derived growth factor C (PDGFC). These CAFs express interleukin 6 (IL6), C-C motif chemokine 2 (CCL2), and Transforming Growth Factor Beta 3 (TGFB3) that induce the differentiation of peripheral blood mononuclear cell (PBMC) to MDSC, which contributes to T cell depletion in the tumor microenvironment [80,81]. CCL2 can also derive from tumor cells undergoing EMT (e.g., through ZEB1), promoting M2 polarization. In turn, polarized M2 macrophages contribute to the EMT of tumor cells through MMP9 [82] and PDL1 expression [83]. Therefore, both CAFs and tumor cells expressing the EMT factors ZEB1, ZEB2, SNAI1, and SNAI2, contribute to the recruitment of MDSC and M2 macrophages inferred in Cluster 1.

Which specific M2 and MDSC cells are present in Cluster 1 and how they are protected from apoptosis are both important questions. Based on transcriptional profiling, we can identify gMDSC as the predominant MDSC in this cluster (Table 1). These specific gMDSC are CCL2-dependent and exert T cell inhibition through ROS/STAT3, but not arginase 1 production [70]. Hydrogen peroxide leads to reduced TCR chain expression and also contributes to TME acidification [71]. Moreover, LDH-A contributes to M2 TAM polarization and TME acidification through lactate and vascular endothelial growth factor (VEGF) expression [84]. The accumulation of extracellular lactate in Cluster 1 would promote an immune permissive microenvironment attenuating dendritic cell [85] and MDSC function through STAT1 activation [86], PD-L1 expression [87], and M2 polarization [88]. Interestingly, PD-L1 expression by STAT1 is activated by type-I IFN in an autocrine manner, suggesting that PD-L1 expression is not related to TIL recognition. This aspect can have important clinical implications because PD-L1 in myeloid cells has been tested as a biomarker of TIL presence, in prospective ICB clinical trials. Finally, CCL2-dependent gMDSC would evade apoptosis through the inhibition of the intrinsic death pathways by the upregulation of MCL-1 [89], which is also overexpressed in Cluster 1.

The above considerations lead us to speculate that Cluster 1 tumors display a hypoxic TME high in lactate that impacts the interactions between tumor cells, fibroblasts, macrophages, and immune cells (Figure 1). In a proportion of these tumors, mutant BRAF may partly explain an overactivation of the Wnt-β-catenin-MYC axis which, along with dysregulated post-translational processes such as PIAS-mediated sumoylation, would contribute to a metabolic reprogramming characterized by enhanced glycolysis in tumor cells, gMDSC, and fibroblasts. The latter would participate in a reverse Warburg effect [90], contributing to a high lactate microenvironment and consequent acidification.

We also speculate that the metabolic reprogramming of tumor cells and the lactate-rich microenvironment dictate the associated immune microenvironment and angiogenesis (Figure 1). In this model, CD4 T*_reg_* [15] and exhausted CD8 T [91] cells (expressing CD39 ectonucleotidase (ENCTD1) are the two main TIL populations in Cluster 1. As such, T*_reg_* cells display high pyruvate dehydrogenase and OXPHOS activities. In this specific scenario, lactate impairs CD8 T cell function, which is strictly glucose-dependent. A TGFB-rich environment, observed in Cluster 1, would also prime CD39 (ENCTD1)-dependent upregulation of CD8 TILS [92]. Importantly, this specific CD8 TIL subpopulation was found in MSI-H colorectal patients, suggesting that not only the tumor mutational burden, but also specific metabolic reprogramming, could be important in CBI efficacy. Tumor-driven inflammation (e.g., mediated by MMP7) [93], TAMS [88,94], and a lactate-rich TME [95]) would support angiogenesis (VEGFA upregulation) in Cluster 1.

#### 6.1.2. Rationale for the Design of Clinical Trials

After chemotherapy and radiotherapy, gastrointestinal tumors can acquire an EMT phenotype similar to that observed in Cluster 1 [96,97]. Additionally, the first-line combination regime 5-FU, oxaliplatin, and bevacizumab (FOLFOX–bevacizumab) causes a shift in the ratio of tumor-associated T cell subpopulations. Therefore, tumor infiltration with T*_reg_*, T_H_17, and gMDSC is significantly increased in mCRC patients treated with this therapy, which can lead to therapeutic resistance. Indeed, an increased T_H_17 and gMDSC frequency is associated with a poor prognosis. In this regard, it is important to note that gMDSCs expressing high levels of PD-L1, CD39, and CD73 exert robust immunosuppressive activity [98]. Although bevacizumab has been shown to reduce the presence of T*_reg_* in the peripheral blood of mCRC patients, the clinical significance of this observation remains to be established [99].

Immune checkpoint-targeted therapies have shown promise in MSI mCRC patients with high TMB, but not in patients with MSS tumors, characteristically with low TMB. However, scant emphasis has been placed on metabolic and TME reprogramming as factors that may limit the capacity of the adaptive immune system to mount a robust anti-tumor response.

Drugs that might reprogram the highly glycolytic metabolism inferred in Cluster 1 are inhibitors of pyruvate dehydrogenase kinase 1 [100], lactate dehydrogenase A [101], or the lactate transporters MCT1 and MCT4 [102]. TAMs and MDSC in the TME may favor the escape of tumor cells from immune surveillance. We speculate that, in Cluster 1, bone marrow-derived CCR2^+^ gMDSC are recruited to the TME by neoplastic cells and CAFs, which would be prevented by CCR2 inhibitors. Small-molecule CCR2 inhibitors (CCR2i) improve gemcitabine’s efficacy and increase CD8 infiltration in pancreatic cancer [103], and increase the overall survival in mice with mCRC liver tumors [104]. In murine models, PD-1 inhibitors plus Ibrutinib [105,106] and PD-1 inhibitors plus anti-TGFβ [107,108] have shown anti-tumor activity in colorectal cancer. As a cautionary note, all these studies used murine-derived pancreatic and colon cancer models, and the extent to which they adequately recapitulate the complexity of human colon cancer is thus unclear. Several ongoing clinical trials are currently testing the efficacy of ICB in combination with antiangiogenic drugs. Given the dependency of angiogenesis and TILs on specific tumor metabolic reprogramming and TME, it appears unlikely that the combination of ICB with antiangiogenic drugs for unselected MSS mCRC patients may prove efficacious.

### 6.2. Cluster 2: Inflamed Non-Stromal Dependent

#### 6.2.1. Metabolism and Microenvironment

GAIS-42 Cluster 2 encompasses 20%–25% of mCRC patients and its transcriptional profile suggests a strong association with immune (T, NK, and B cells) and myeloid-monocytic cells with enrichment for immune checkpoint genes and features of EMT (Table 1). This cluster appears to have undergone a distinctive metabolic reprogramming with enhanced TCA oxidative phosphorylation and glutaminolysis (high levels of GLS) (Figure 1) [109]. Glutaminolysis in tumor cells can be MYC-dependent and feeds mitochondrial metabolism through TCA cycle anaplerosis [110]. Glutamine dehydrogenase (GDH) and glutamine synthetase (GS) are also upregulated in Cluster 2, which may allow glutamine biosynthesis from glutamate and ammonia with a lower reliance on glutamine [111]. Cluster 2 tumors are inferred to replenish glutamate not only through GLS1, but also through the branched-chain amino acid aminotransferase, BCAT1 [112]. In mCRC, MYC overexpression can be a consequence of mutations in FBXW7, a component of the SCF-like ubiquitin ligase complex, which lead to an accumulation of its target proteins, including MYC and p53. FBXW7 mutations are observed in roughly 8%–10% of mCRC and may contribute to chromosomal instability and angiogenic resistance [113,114], as well as to a shift of cellular metabolism toward oxidative phosphorylation [115]. Strong tumor glutaminolysis, such as that inferred for Cluster 2 tumors, compromises the availability of glutamine for other cellular populations in the TME (Figure 1). For example, in addition to glycolysis, glutaminolysis is a major source of energy production required by activated T cells and is also dependent on MYC, as the deletion of MYC profoundly inhibits the upregulation of glutamine oxidation in active T cells [116]. ASCT2, a glutamine transporter, is crucial for mediating TCR-stimulated glutamine uptake during the differentiation of naive CD4 T cells, but not for the generation of T*_reg_* [117]. Finally, glutamine deprivation promotes Foxp3 T*_reg_* [26] and inhibits the proliferation of both T_H_1 and T_H_17 cells [27].

MDSC and TAMs are also inferred to contribute to T cell dysfunction in the inflamed Cluster 2, possibly independently of fibroblasts. Cluster 2 shows less fibroblast dependency than Cluster 1, although the EMT factor TWIST and its transcriptional target CXCL12 are clearly upregulated in this cluster [118]. CXCL12 has been associated with CD8 depletion and resistance to anti-PD-1 in pancreatic cancer [119]. At least three major pathways could contribute to the TME reprogramming and the significant presence of T*_reg_* inferred for Cluster 2. First, activated inflammatory dendritic cells (inf-DC) produce IL-23 and select for γδT-17 cells, which secrete IL-17 and facilitate the development of T*_reg_* [120]. Inf-DCs promote gMDSC as an outcome of arginine depletion through arginase 1 [121], which is also over-expressed in Cluster 2, further contributing to an immunosuppressive tumor environment. gMDSCs can evade apoptosis through FAS downregulation, as is the case in Cluster 2 [122]. Secondly, angiopoietin 2, a proangiogenic cytokine expressed by endothelial cells, promotes a highly immunosuppressive TIE2-expressing macrophage subpopulation that enables the accumulation of T*_reg_* through the secretion of IL-10 [123] (also upregulated in Cluster 2). We speculate that these specific TAMs express PD-1 (also upregulated in Cluster 2), which limits the anti-tumor efficacy of anti-PD-1 CBI [124]. Thirdly, tumor-derived TGFB1 (upregulated in Cluster 2) induces CD39 γδT*_reg_* cells in colorectal cancer [125], also potentially contributing to an immunosuppressed TME in Cluster 2 (Figure 1). 

Of interest in Cluster 2 is the upregulation of IL-2 and CD8 and the downregulation of ENTPD1 (CD39). Recently, a subpopulation of bystander CD8^+^ T cells that lacked CD39 and were activated by non-tumoral antigens was described in colorectal cancer [126]. In addition, IL-2 is essential for the suppression of CD8 T cells by T*_regs_* [127]. Based on the observed immune-phenotype, we propose that T*_reg_* associated with Cluster 2 tumors are characterized by IL-10 and TGFB1 overexpression, possibly induced by mutant KRAS tumor cells [125,128] or by endothelial cells through angiopoietin 2 [123]. Recently, different populations of T*_reg_* have been described in colorectal cancer, based on the expression of FOXP3 [129] and CD25 [130]. The Cluster 2 profile suggests that angiogenesis can be engaged through IL-17 [131], IL-10, and prostaglandin E2 [132], thus following a mechanism distinct from Cluster 1. These cytokines induce the expression of the Fas ligand (FASL) in endothelial cells and may deplete CD8^+^ T cells, but not T*_reg_*, contributing to the immune suppressive environment associated with Cluster 2 [133].

#### 6.2.2. Rationale for the Design of Clinical Trials

In untreated MSS patients, Cluster 2 constitutes only 20% of all mCRC. However, heavily pre-treated mCRC patients [134]) with conventional therapies such as bevacizumab, which causes CXCR4 upregulation [135] or cetuximab, which induces IL-10-dependent M2 polarization of TAMs, could potentially convert tumors initially identified as Cluster 1 or 3 into Cluster 2-like tumors. Drugs that could target the high glutaminolytic metabolism inferred for Cluster 2 are inhibitors of glutaminase (e.g., CB-839), pyruvate carboxylase, or both, and drugs that inhibit CXCR4 [136] or arginase 1 such as CB-1158 [137], all of which are expected to potentiate the efficacy of ICB.

### 6.3. Cluster 3: Non-Inflamed/Cold

#### 6.3.1. Metabolism and Microenvironment

GAIS-42 Cluster 3 encompasses 40% of patients and is characterized by low levels of the expression of genes expressed in GAIS-42 Cluster 1 and Cluster 2. This cluster correlates with IO360 Cluster 3 (*p* < 0.001).

Tumors in Cluster 3 are inferred to display a metabolic profile similar to the Warburg-type glycolytic Cluster 1 (with upregulation of PKM2, LDH-A, and PDK1) (Figure 1). In contrast, it is also characterized by a lack of evidence of an association with fibroblasts, macrophages with M2 polarization, MDSCs, and T*_reg_* (Table 1). An additional feature of Cluster 3 is an increased expression of hexokinase 2 (HK2) that can also promote glucose consumption and downstream lactate production [138]. Glycolytic tumor cell metabolism limits glucose availability in the TME which, coupled with a high lactate excretion and extracellular acidification rate (ECAR), can potentially dampen CD8 T cell differentiation and function [14]. As such, activated T cells engage aerobic glycolysis and anabolic metabolism for growth, proliferation, and effector functions. A glucose-poor tumor microenvironment limits aerobic glycolysis in tumor-infiltrating T cells, which suppresses their tumoricidal effector functions. Although the glycolytic metabolite phosphoenolpyruvate (PEP) sustains T cell receptor-mediated Ca(2+)-NFAT signaling and effector functions by repressing sarco/ER Ca(2+)-ATPase (SERCA) activity, it promotes CD8 T cell dysfunction in tumors with a high glucose consumption [139], such as those in Cluster 3. In addition, iNOS2 produced by non-polarized macrophages (M1) [140] in conjunction with HIF-1α coordinates DC to limit DC-stimulated T-cell responses [141].

We speculate that in the presence of low CD8^+^ T cell infiltration (non-antigen-targeting TILs; e.g., MSS tumors), T cells do not compete with activated DC for glucose consumption and the mTORC1/HIF1/iNOS2 signaling axis thus represses DC-induced T cell responses. When antigen presentation is boosted (e.g., MSS tumors after vaccine priming), tumors might inhibit immunity, despite high CD8^+^ infiltration, through glucose consumption and TIL TCR impairment.

#### 6.3.2. Rationale for the Design of Clinical Trials

From the above description, we postulate that Cluster 3 tumors are potentially sensitive to CBI therapy. PD-1 and PD-L1 inhibitors can down-regulate PD-L1 and decrease ECAR and tumor glycolysis [14]. In MSS mCRC patients, oncolytic virotherapy, specific neoepitope vaccines or autologous or heterologous vaccines associated with CBI, can potentially be active [142,143], although, currently, there is no evidence that these interventions result in CD8^+^ T cell increments similar to MSI. In spite of the glucose requirement for effector T cell differentiation and function, it is likely that strategies to limit the glycolytic activity (which would impact both tumor and immune cells) would boost CBI activity. HK2 is upregulated in about 70% of patients in Cluster 3. Targeting HK2 and glycoytic dependency would result in the depletion of CD276^+^ tumor cells and inhibition of neoangiogenesis [144]. Because a high glycolytic consumption by tumor cells can dampen T-cell activation through NFAT1 down-regulation, interventions that have been shown to up-regulate NFAT1 in T-cells, including CDK4/6 inhibitors, such as amebaciclib or palbociclib, would also merit consideration. CDK4/6 inhibitors have demonstrated activity in combination with CBI in breast and colorectal cancer preclinical models [145,146]. However, CDK4/6 inhibition can increase MYC-glutamine dependence and thus acquisition of resistance due to metabolic reprogramming [147]. Therefore, glutaminase inhibitors such as CB-839, in combination with CDK4/6, with or without ICB, would be a potentially useful therapeutic strategy for Cluster 3 patients.

## 7. Conclusions

ICB has proven to be beneficial to MSI mCRC patients. However, the majority of mCRC patients are MSS and do not benefit from these therapies. Classical phase II or phase III clinical trials in non-selected mCRC MSS patients with ICB alone [32] or combinations with MEK inhibitors [148] have been shown to be ineffective. We propose that trial design should be based on specific immune and metabolic characteristics associated with properly classified tumors. Our immune-metabolic clusters provide a basis for the design of potentially efficacious drug combinations. Clusters 1 and 2 could benefit from drugs that modify the metabolic reprogramming of tumor cells and the TME. Cluster 3 patients could benefit from interventions that increase antigen presentation (pe. oncolytic virotherapy, specific neoepitope vaccines, or autologous or heterologous vaccines) in combination with ICB and drugs that modify glycolytic reprogramming. We advocate for biomarker-driven phase II trials with combinations of immune and metabolic modifiers, in order to improve the therapeutic efficacy in MSS mCRC.

## Figures and Tables

**Figure 1 cancers-11-01172-f001:**
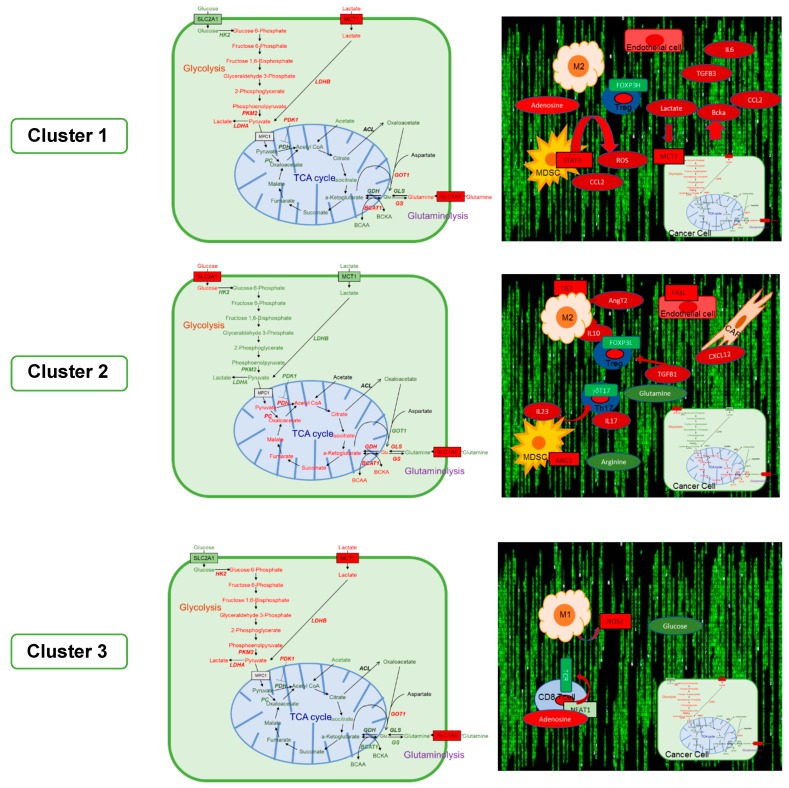
Major metabolic modules and tumor microenvironment interactions inferred for the GAstrointestinal Immune-Signature (GAIS)-42 immune clusters. Left panels. Major cluster-specific metabolic modules inferred for tumor cells are highlighted in red (up-regulated) or green (down-regulated). Differential expression of corresponding enzyme/metabolite is indicated by color (red, up-regulated; green, down-regulated). Right panels. Interactions between tumor cells, stromal fibroblasts, and tumor-infiltrating lymphocytes (TILs). Round nodes represent metabolites and square nodes represent enzymes. Abbreviations: ACL, ATP-citrate synthase; ANGT2, angiopoietin 1; ARG1, arginase 1; BCAA, branched-chain-amino-acid aminotransferase; BCAT1, branched-chain-amino-acid aminotransferase, cytosolic; BCKA, branched-chain ketoacids; CAF, cancer-associated fibroblasts; CCL2, C-C motif chemokine 2; CD8, T-cell surface glycoprotein CD8 alpha; CXCL12, C-X-C Motif Chemokine Ligand 12; FASL, Fas ligand; FOXP3, forkhead box P3; GLS, glutaminase; GOT1, glutamic-oxaloacetic transaminase 1; GS, glutamate-ammonia ligase; HK2, hexokinase 2; IL10, interleukin 10; IL17, interleukin 17; IL23, interleukin 23; IL6, interleukin 6; LDHA, lactate dehydrogenase A; LDHB, lactate dehydrogenase B; M1, M1-type macrophages; M2, M2-type macrophages; MCT1, monocarboxylate transporter 1 (official gene symbol SLC16A1); MDSC, myeloid-derived suppressor cells; iNOS2, nitric oxide synthase 2 inducible; PC, pyruvate carboxylase; PDH, pyruvate dehydrogenase; PDK1, pyruvate dehydrogenase kinase 1; PKM2, pyruvate kinase muscle isozyme 2; ROS, reactive oxygen species; SLC1A5, solute carrier family 1 member 5 (glutamine transporter); SLC2A1, solute carrier family 2 member 1 (glucose transporter); STAT3, signal transducer and activator of transcription 3; TCR, T cell receptor; TGFB1, Transforming Growth Factor Beta 1; TGFB3, Transforming Growth Factor Beta 3; TIE2, tyrosine-protein kinase receptor; T_*reg*_, regulatory T cells; TH17, type-17 helper T cells.

**Table 1 cancers-11-01172-t001:** Biologically and clinically relevant features in metastatic colorectal cancer (mCRC) as inferred for GAstrointestinal Immune-Signature (GAIS)42 immune-metabolic (IM) clusters.

Features	Cluster 1	Cluster 2	Cluster 3
	Inflamed-Stromal Dependent	Inflamed-Non Stromal Dependent	Non-Inflamed Cold
BRAF, RAS Mutations	BRAF	RAS	none
EMT	+++	+	-
Angiogenesis	+++	+++	+
CAF	+++	-	-
MDSC/M	M2 (pSTAT3) PMF-MDSC (ROS)	M2 (ARG1) PMF-MDSC (ARG1)	M1 (iNOS2)
TILs	T_*reg*_+++ (CD39^+^CD25^−^)	T_*reg*_+++ (CD39^−^ CD25^+^). T_H_2-like T_reg,_ γδT17	T_H_17, T_H_1
Metabolism in Tumoral Cells	Glycolytic (LDHA, PDK1, PKM2, MCT1)	Glutaminolytic and OXPHOS (GLS1, PC, GLUD1)	Glycolytic (LDHA, PDK1, PKM2, MCT1, HK2)
Extracellular Metabolites	High lactate High adenosine High BCKAs	Low glutamine Low arginine	Low glucose High adenosine
Potential Therapeutic Targets	*PD-1, PI3Kδ, STAT3, BTK* *LDH, PDK1, MCT1*	*PD-1, ARG1, BTK, CCR5* *GLS1, PC*	*PD-1, CDK4/6* *PDK1, MCT1* *HLDCV, TVEC*

Abbreviations: ARG1, arginase 1; BCKA, branched chain ketoacids; BTK, Bruton’s tyrosine kinase; CAF, cancer-associated fibroblasts; CCR5, chemokine receptor 5; CDK4/6, cycline-dependent kinases 4 and 6; EMT, epithelial-mesenchymal transition; GLS1, glutaminase 1; GLUD, glutamate dehydrogenase; HK2, hexokinase 2; HLDCV, heterologous dendritic cell vaccines; LDHA, lactate dehydrogenase A; M1, M1-type macrophages; M2, M2-type macrophages; MCT1, monocarboxylate transporter 1 (official gene symbol SLC16A1); MDSC, myeloid-derived suppressor cells; iNOS2, nitric oxide synthase 2 inducible; OXPHOS, oxidative phosphorylation; PC, pyruvate carboxylase; PD-1, programmed cell death 1; PDK1, pyruvate dehydrogenase kinase 1; PI3Kδ, phosphatidylinositol-3 kinases; PKM2, pyruvate kinase muscle isozyme 2; PMF, polymorphonuclear; ROS, reactive oxygen species; STAT3, signal transducer and activator of transcription 3; T_*reg*_, regulatory T cells, T_H_1, type-1 helper T cells; T_H_2, type-2 helper T cells; T_H_17, type-17 helper T cells; TIL, tumor infiltrating lymphocytes; TVEC, Talimogene laherparepvec. – absent, + present, +++ abundant

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
