# Peer review of "The Tumor Microenvironment in Colorectal Cancer Therapy"

_cancers, 2019, doi:10.3390/cancers11081172_

Round 1

Reviewer 1 Report

In the manuscript entitled “Tumor Microenvironment in Colorectal Cancer Therapy” Pedrosa and others have focused on mCRC cancer and have described the classification of mCRC into three immune-metabolic clusters. They emphasize on the fact that designing clinical trails should consider immune and metabolic properties of different mCRC clusters and suggested designing treatment strategies based on which cluster a patient falls in. In general the review is written and described well. My minor comments that the authors may want to consider are:

1.    The authors started explaining mCRC first (line 30/53) and then moved to describing cancers in general (line 87/136) and then again started describing CRC/mCRC (line 179). This may be distracting to the readers so the authors could start with a general description about cancers first and then fully focus on CRC/mCRC.

2.    The general readers who are not very familiar with this topic may want to see an opening paragraph describing about each title. For e.g. line 260- an opening paragraph about what does it mean by "inflamed-stromal dependent" would be helpful. Likewise for other titles.

3.    The font size for figure 1 is too small to read. The authors either could enlarge the size or color code it with numbering and describe the numbering in the captions. It would also be helpful if the authors could mention the cluster no. above the figure.

4.    Tyops – line 53 and 87 title no. 2 has been repeated, spell error eg tumour/tumor (line 201/209), -ketoglutarat (line 286), PM-MDSCand (line 239), anti/PD1 (line 409). 

Author Response

Dear Ms/Mrs.

We will response point by-point your commnets:

1.    The authors started explaining mCRC first (line 30/53) and then moved to describing cancers in general (line 87/136) and then again started describing CRC/mCRC (line 179). This may be distracting to the readers so the authors could start with a general description about cancers first and then fully focus on CRC/mCRC.

In agreement with the comment reviewer, we have now started with a general description of cancer first and then focus on CRC/mCRC.

2.    The general readers who are not very familiar with this topic may want to see an opening paragraph describing about each title. For e.g. line 260- an opening paragraph about what does it mean by "inflamed-stromal dependent" would be helpful. Likewise for other titles.

We agree with the reviewer. We have explained in brief what it means each cluster subtype.

3.    The font size for figure 1 is too small to read. The authors either could enlarge the size or color code it with numbering and describe the numbering in the captions. It would also be helpful if the authors could mention the cluster no. above the figure.

We have changed figure 1 significantly. We hope that the new version will be clearer. In addition, we have defined the abbreviations.

4.    Tyops – line 53 and 87 title no. 2 has been repeated, spell error eg tumour/tumor (line 201/209),
a-ketoglutarat (line 286), PM-MDSCand (line 239), anti/PD1 (line 409).

We have corrected all Typos.

Reviewer 2 Report

The authors provide a comprehensive analyses of the current limitations of therapies for mCRC and propose alternative approaches to treat MSS mCRC patients along with the rationale for utilizing such approaches. Current therapies do not select for patients based on their tumor immune compartments and metabolic statuses of the TME contributing to lack of efficacy. In this scenario, it is pivotal to arrive at a classification system that encompasses tumor and tumor-associated stroma and TIL phenotypes. The authors propose the use of GAIS-42 classification that divides mCRC tumors into 3 clusters depending on their metabolic status, MDSC/ Macrophage and CAF compartments and angiogenic dependence. 

The review is well written with sufficient background on mCRC, the tumor immune infiltration and metabolic dependencies of tumor and different T cell compartments. The "rationale of clinical trial design" portion after the explanation of each cluster is especially well written. The review is easy to follow and understand, but English changes need to be made at certain areas to make it clearer. There were certain lines that were unclear and confusing and are mentioned below.

1) Line 45- The authors use the term "immune checkpoint blockade inhibitors" which is not correct. The term needs to be "immune checkpoint blockade" which already means the immune checkpoints are being inhibited. This can be abbreviated as ICB and used throughout the text.

2) Line 196-198The quantification of T cells and cytotoxic T cells (CD3 and CD8) in mCRC tumors (Immunoscore) correlates with a decreased likelihood of metastasis (Mlecnik, 2016) and can also predict overall survival (Mlecnik, 2018). I believe the authors mean higher immunoscore correlates with decreased likelihood of metastasis. Is that correct? This line needs clarification.

3) Line 317-318Accumulation of extracellular lactate in Cluster 1 would promote an immune permissive microenvironment attenuating dendritic cell (Gottfried, 2006) and MDSC function through STAT1 activation. Lactate accumulation should create an immune suppressive environment with attenuated DC and enhanced MDSC function. Clarify this statement. 

4) Line 341-342: A TGFB3 rich environment, observed in cluster 1, would also prime CD39 (ENCTD1)-dependent upregulation of CD8 TILS. Does this mean TGFb3 upregulates CD39 on CD8 TIL? Also briefly mention the importance of this particular CD8 population.  

Author Response

Dear Mr/Ms.

We will response the comments point by-point:

1) Line 45- The authors use the term "immune checkpoint blockade inhibitors" which is not correct. The term needs to be "immune checkpoint blockade" which already means the immune checkpoints are being inhibited. This can be abbreviated as ICB and used throughout the text.

In agreement with the reviewer’s comment, we have changed the term "immune checkpoint blockade inhibitors" for immune checkpoint blockade" and the abbreviated ICB was used thereafter.    

2) Line 196-198: The quantification of T cells and cytotoxic T cells (CD3 and CD8) in mCRC tumors (Immunoscore) correlates with a decreased likelihood of metastasis (Mlecnik, 2016) and can also predict overall survival (Mlecnik, 2018). I believe the authors mean higher immunoscore correlates with decreased likelihood of metastasis. Is that correct? This line needs clarification.

This is correct. We have re-written this paragraph.

3) Line 317-318: Accumulation of extracellular lactate in Cluster 1 would promote an immune permissive microenvironment attenuating dendritic cell (Gottfried, 2006) and MDSC function through STAT1 activation. Lactate accumulation should create an immune suppressive environment with attenuated DC and enhanced MDSC function. Clarify this statement. 

This is a very interesting point. In the papers published by Xiao and Lu, PD-L1 in MDSC was upregulated by STAT1 in through autocrine INF-g interferon independent mechanism. This aspect can have important clinical implications because PD-L1 in myeloid cells has been tested as a biomarker of TIL presence, in prospective ICB clinical trials. We have added this aspect in the modified version of the manuscript.

4) Line 341-342: A TGFB3 rich environment, observed in cluster 1, would also prime CD39 (ENCTD1)-dependent upregulation of CD8 TILS. Does this mean TGFb3 upregulates CD39 on CD8 TIL? Also briefly mention the importance of this particular CD8 population.

This is only a hypothesis, because TGFb3 and ENCTD1 were over-expressed in cluster 1. The manuscript by Duhen et al. suggests that TGF-b up-regulate CD39. In accordance we have changed TGFb3 for TGFb in the modified manuscript.

Thank you so much.